# Prevalence and Influencing Factors of Metabolic Syndrome among Adults in China from 2015 to 2017

**DOI:** 10.3390/nu13124475

**Published:** 2021-12-15

**Authors:** Fan Yao, Yacong Bo, Liyun Zhao, Yaru Li, Lahong Ju, Hongyun Fang, Wei Piao, Dongmei Yu, Xiangqian Lao

**Affiliations:** 1Key Laboratory of Trace Element Nutrition of National Health Commission, National Institute for Nutrition and Health, Chinese Center for Disease Control and Prevention, Beijing 100050, China; yao_f@foxmail.com (F.Y.); zhaoly@ninh.chinacdc.cn (L.Z.); julh@ninh.chinacdc.cn (L.J.); fanghy@ninh.chinacdc.cn (H.F.); piaowei@ninh.chinacdc.cn (W.P.); 2Jockey Club School of Public Health and Primary Care, The Chinese University of Hong Kong, Hong Kong 999077, China; boyacong777@gmail.com (Y.B.); xqlao@cuhk.edu.hk (X.L.); 3Beijing Friendship Hospital, Beijing 100050, China; liyaru0225@126.com

**Keywords:** metabolic syndrome, adults, prevalence, influencing factors

## Abstract

The prevalence and influencing factors of metabolic syndrome (MetS) in Chinese residents aged 20 or older were investigated. The data were collected from China Nutrition and Health Surveillance (2015–2017), which used a stratified, multistage, random sampling method. A total of 130,018 residents aged 20 years or older from 31 provinces were included in this study. The National Cholesterol Education Programme Adult Treatment Panel III (NCEP ATP III) criteria were used to define MetS. The standardised prevalence of high waist circumference, high blood pressure and low high-density lipoprotein cholesterol were 40.8%, 49.4% and 41.1%, respectively. The following factors were associated with a higher prevalence of MetS: female [odds ratio (OR) = 1.773, 95% CI = 1.709–1.840]; older age (OR = 1.037, 95% CI = 1.036–1.039); living in north China (OR = 1.087, 95% CI = 1.058–1.117); high body mass index (OR = 1.402, 95% CI = 1.395–1.408); higher income [OR (95% CI): 1.044 (1.007–1.083), 1.083 (1.044–1.124) and 1.123 (1.078–1.170) for moderate, high, and very high income, respectively]; family history of hypertension (OR = 1.237, 95% CI = 1.203–1.273); family history of diabetes (OR = 1.491, 95% CI = 1.426–1.558) and current smoking status (OR = 1.143, 95% CI = 1.098–1.191). Living in the countryside (OR = 0.960, 95% CI = 0.932–0.988), moderate alcohol consumption (OR = 0.917, 95% CI = 0.889–0.946) and being physically active (OR = 0.887, 95% CI = 0.862–0.913) were associated with a lower prevalence of MetS. The prevalence of MetS among residents aged 20 years or older in China is increasing, especially among women, people aged 45 years or older and urban residents. Preventive efforts, such as quitting smoking and engaging in physical activity, are recommended to reduce the risk of MetS.

## 1. Introduction

Metabolic syndrome (MetS) refers to a set of combined cardiovascular risk factors that, together, constitute cardiovascular risk beyond the sum of the individual components. The main components of MetS are abdominal obesity, insulin resistance, increased blood pressure and dyslipidaemia [1]. With economic development and changes in lifestyle, the prevalence of MetS is increasing worldwide [2,3] and has become a public health issue owing to its serious effect on human health. In 2001, a nationally representative sample survey conducted in 31 provinces and cities in China found that the standardised prevalence of MetS was 13.7% (men 9.8%, women 17.8%) [4]. In 2010–2012, the standardised prevalence of MetS was found to have increased to 24.2% (men 24.6%, women 23.8%) [5]. These data indicate the urgent need for the prevention and control of MetS. Based on data from China Nutrition and Health Surveillance (2015–2017), this study evaluated the prevalence and influencing factors of MetS among Chinese residents aged 20 years or older.

## 2. Materials and Methods

### 2.1. Data Source

The data were derived from China Nutrition and Health Surveillance (2015–2017), which adopted a stratified, multistage, random sampling method to recruit representative participants from 31 provinces/municipalities/autonomous regions of China. Detailed information about this survey is given in our previous report [6]. The protocol for the current study was approved by the Ethics Committee of the National Institute for Nutrition and Health, and the Chinese Centre for Disease Control and Prevention (approval numbers: 201519-A and 201614), and all of the participants signed an informed consent form prior to joining the study. A total of 130,018 participants aged 20 years or older were included in the analysis.

### 2.2. Data Collection

China Nutrition and Health Surveillance (2015–2017) collected information using the four following parts: a questionnaire survey, a physical examination, a dietary evaluation and a laboratory test.

#### 2.2.1. Questionnaire Survey

A face-to-face questionnaire was used to collected family and personal information, including information on lifestyle factors, health status and physical activity. 

#### 2.2.2. Physical Examination

Height, weight, waist circumference and blood pressure were measured by trained investigators based on standard methods using a TZG height and sitting height meter, a TANITA HD-390 electronic weighing scale, a waist circumference ruler and an Omron HBP1300 electronic sphygmomanometer. These measurements were accurate to 0.1 cm, 0.1 kg, 0.1 cm and 1 mmHg, respectively.

#### 2.2.3. Dietary Evaluation

The food frequency questionnaire was used to capture and evaluate the dietary structure and dietary habits of participants over the past year. Specifically, information was collected regarding the consumption of vegetables, fruits and red meat items.

#### 2.2.4. Laboratory Tests

An overnight fasting blood sample was collected from each participant to measure blood biochemical indexes and nutritional status parameters.

### 2.3. Quality Control

To ensure quality, the National Project Working Group devised a quality control plan and supervised its implementation. This ensured that there were unified programmes, manuals and questionnaires; unified training and assessment; unified equipment and reagents; and unified data entry and data cleaning across the study sites.

### 2.4. Definition of MetS

MetS was defined according to the National Cholesterol Education Programme-Adult Treatment Panel III (NCEP-ATP III) [7] as having three or more of the following factors: (1) a waist circumference of ≥90 cm for men and ≥80 cm for women; (2) a systolic blood pressure of ≥130 mmHg or a diastolic blood pressure of ≥85 mmHg or receiving anti-hypertension treatment; (3) a fasting triglyceride level of ≥1.7 mmol/L or receiving corresponding treatment; (4) a high-density lipoprotein cholesterol (HDL-C) level of <1.03 mmol/L for men and <1.30 mmol/L for women or receiving corresponding treatment; (5) a fasting plasma glucose (FPG) level of ≥5.6 mmol/L or receiving anti-diabetes treatment or reporting previously physician-diagnosed diabetes.

### 2.5. Covariates

A wide range of potential confounders were accounted for: (1) body mass index (BMI) was categorised as normal (18.5 ≤ to < 24 kg/m^2^), overweight (24 ≤ to < 28 kg/m^2^) or obese (≥28 kg/m^2^); (2) education level was categorised as low (primary school or below), moderate (junior school) or high (high school or above); (3) according to the income quartile, income was categorised as low, moderate, high or very high; (4) family history of hypertension (or diabetes) was defined as one or more of a grandfather, grandmother, father, mother or brother/sister suffering from hypertension (or diabetes); (5) smoking was categorised as never smoked, formerly smoked or currently smoke; (6) alcohol consumption was categorised as never consumed, moderately consume (men consume less than 25 g of alcohol and women less than 15 g of alcohol per day) or excessively consume (men consume more than 25 g and women more than 15 g per day); (7) physical activity was defined as inactive if, within one week, the total time of moderate-intensity activity was less than 150 min, or high-intensity activity was less than 75 min, or the cumulative amount of moderate- and high-intensity activity was less than 150 min [8]; (8) fruit and vegetable intake was defined as insufficient if the daily average intake was <400 g [9]; (9) red meat intake was categorised as insufficient (daily average red meat intake <18 g), moderate (18 g ≤ to < 27 g) or excessive (≥27 g) [10].

### 2.6. Statistical Analysis

SAS 9.4 software (SAS Institute Inc., Cary, NC, USA) was used to analyse the data. Continuous variables with normal distributions are presented as $\bar{x}$ ± s, and comparisons between groups were made using the *t*-test. Variables with skewed distributions are presented as the median with the 25th and 75th quantiles [M (P25, P75)], and comparisons between groups were made using the non-parametric statistical hypothesis test. Categorical variables are expressed as N (%) and were compared by the chi-square test. The PROC SURVEYFREQ was used to calculate the standardised prevalence and 95% CI of MetS and its components using the weight derived from the data published by the China National Bureau of Statistics in 2010. The influencing factors were analysed using a logistic regression model. A two-sided *p* value < 0.05 was considered to indicate statistical significance.

## 3. Results

### 3.1. General Characteristics of the Participants

A total of 130,018 participants were included—61,775 men (47.5%) and 68,243 women (52.5%). The average age of the men in the study was higher than that of the women. There were significant differences in MetS prevalence by sex, age, urban vs. rural residence, area of the country, education level, income, family history of hypertension, family history of diabetes, smoking status, alcohol consumption, physical activity, fruit and vegetable intake and red meat intake. Waist circumference, systolic blood pressure, diastolic blood pressure, triglycerides and FPG were significantly higher in men than in women, while BMI and high-density lipoprotein were significantly lower in men than in women (*p* < 0.05, Table 1).

### 3.2. Standardised Prevalence of MetS and Its Components

The standardised prevalence of MetS among Chinese residents aged 20 years or older was 31.1%, with a significantly higher prevalence in women than in men (32.3% vs. 30.0%). The highest prevalence was found in participants aged ≥ 75 years (44.2%), and the lowest prevalence was found in participants aged 20–44 years (23.3%). The prevalence in the north (35.9%) was higher than that in the south (27.4%). There were significant differences in the prevalence of MetS among the groups of BMI, education level, family history of hypertension, family history of diabetes, physical activity, smoking, alcohol consumption and red meat intake (*p* < 0.05, Table 2).

The standardised prevalences of high WC, high BP, elevated TG, low HDL-C and elevated FPG were 40.8%, 49.4%, 29.3%, 41.1% and 24.6%, respectively. Among them, the standardised prevalences of high WC, high BP and low HDL-C were the highest (Figure 1). Among the 130,018 participants, 18,183 (14.0%) had no abnormal components; 32,821 (25.2%) had one abnormal component; 32,136 (24.7%) had two abnormal components; 24,656 (19.0%) had three abnormal components; 15,856 (12.2%) had four abnormal components; and 6366 (4.9%) had five abnormal components (Figure 2).

### 3.3. Multivariate Logistic Regression

Variables with statistical significance in the univariate analysis and those that were not significant but are closely related to MetS according to medical theory were included in the multivariate logistic regression. A stepwise regression method was adopted, and the variables that were finally included in the model were sex, age, location of residence, area of the country, BMI, education level, income, family history of hypertension, family history of diabetes, physical activity, smoking status, alcohol consumption, fruit and vegetable intake, and red meat intake.

The prevalence of MetS was higher in women than in men (OR = 1.773, 95% CI = 1.709–1.840). Every 1 year increase in age was associated with a 3.7% (95% CI = 3.6%–3.9%) increase in the prevalence of MetS. The following factors were associated with a higher prevalence of MetS: living in northern China (OR = 1.087, 95% CI = 1.058–1.117), high BMI (OR = 1.402, 95% CI = 1.395–1.408), higher income [OR (95% CI): 1.044 (1.007–1.083), 1.083 (1.044–1.124), and 1.123 (1.078–1.170) for moderate, high, and very high income, respectively], family history of hypertension (OR = 1.237, 95% CI = 1.203–1.273), family history of diabetes (OR = 1.491, 95% CI = 1.426–1.558) and current smoking (OR = 1.143, 95% CI = 1.098–1.191). Living in the countryside (OR = 0.960, 95% CI = 0.932–0.988), moderate alcohol consumption (OR = 0.917, 95% CI = 0.889–0.946) and being physically active (OR = 0.887, 95% CI = 0.862–0.913) were associated with a lower prevalence of MetS (Table 3).

## 4. Discussion

Based on data from China Nutrition and Health Surveillance (2015–2017), this analysis provides up-to-date information on the prevalence of and risk factors for MetS in Chinese adults. We found that the standardised prevalence of MetS was 31.1%. There was a significantly higher prevalence in women than in men (32.3% vs. 30.0%, *p* < 0.001) and in those living in northern China vs. southern China (35.9% vs. 27.4%, *p* < 0.001). The standardised prevalences of high WC, high BP and low HDL-C were 40.8%, 49.4% and 41.1%, respectively. These three variables are important components of MetS and should be addressed as key factors for MetS prevention and treatment. Only 14.0% of adults had no abnormal MetS components; thus, greater than 4/5 of adults had at least one abnormal MetS component.

Understanding which factors are associated with a high MetS rate would help to identify individuals who are at a greater risk of MetS. Consistent with previous studies, we found that the prevalence of MetS among Chinese adults was related to one’s socioeconomic status and lifestyle factors. Sex, age, location of residence (city or countryside), area of the country (north or south), BMI, income, family history of hypertension, family history of diabetes, smoking status, alcohol consumption and physical activity were related to MetS prevalence. There was a higher prevalence of MetS in women than in men; this trend has also been observed in Porto (24.9% vs. 17.4%) [11], Iran (47.1% vs. 36.5%) [12] and China (15.57% vs. 12.31%) [13]. Postmenopausal status is associated with an increased risk of central obesity and insulin resistance [14], which may be the cause of the sex differences seen in MetS prevalence; however, further research is required to verify this. The MetS prevalence tended to increase with increasing age, which was similar to the findings of Wang et al. [15]. Living in the countryside was associated with a lower prevalence of MetS than living in a city, which was similar to the findings of Xi et al. [16], while living in northern China was associated with a higher prevalence of MetS than living in southern China, which was also observed by Gu et al. [4]. The BMI is an important measure of obesity and overall health and is closely related to MetS [17,18]. This study found that every 1 unit increase in BMI was associated with a 40.2% increase in the prevalence of MetS. No association was found between education level and MetS prevalence, which was consistent with the findings of Park et al. [19]. Participants with higher incomes were more likely to have MetS than those with a low income, which may be related to the influence of economic development on diet structure and health awareness. Prasad et al. found that a middle-to-high socioeconomic status significantly contributed to an increased risk of MetS [20]. A family history of diabetes or hypertension was also significantly related to the prevalence of MetS. Carmelli et al. found that the consistency rates of diabetes, hypertension and obesity in identical twins were significantly higher than those in fraternal twins, indicating that genetic factors have an important relationship with the occurrence of MetS [21]. Current smoking status was a risk factor for MetS, which was consistent with previous studies [22,23]. Compared with never consumed alcohol, moderate alcohol consumption led to a lower risk of developing MetS, and this was consistent with the findings of Tresserra-Rimbau [24]. Being physically active was associated with a lower prevalence of MetS, which was also consistent with recent research showing that exercise can reduce the incidence of MetS [25,26]. Studies have confirmed that red meat intake is positively correlated with MetS occurrence [27,28], while eating more fruits and vegetables is protective against MetS [29,30]. This study did not identify any correlations between diet and the risk of MetS. This lack of correlation may have been related to bias. Furthermore, individuals who had previously developed MetS or a component of MetS may have changed their diet.

The survey data are representative of the whole country. After weighted adjustment, this study well reflects the epidemic characteristics of MetS among residents aged 20 years and older in China. However, this study also has some limitations. First, this was a cross-sectional study; thus, causal relationships could not be determined. Second, the investigation of alcohol consumption was limited to individuals’ consumption patterns within the previous 12 months; therefore, alcohol consumption may have been underestimated.

## 5. Conclusions

We found that the prevalence of MetS among residents aged 20 years or older in China is increasing, especially among women, those aged 45 years or older and urban residents. Preventive efforts, such as quitting smoking and engaging in physical activity, are recommended to reduce the risk of MetS.

## Figures and Tables

**Figure 1 nutrients-13-04475-f001:**
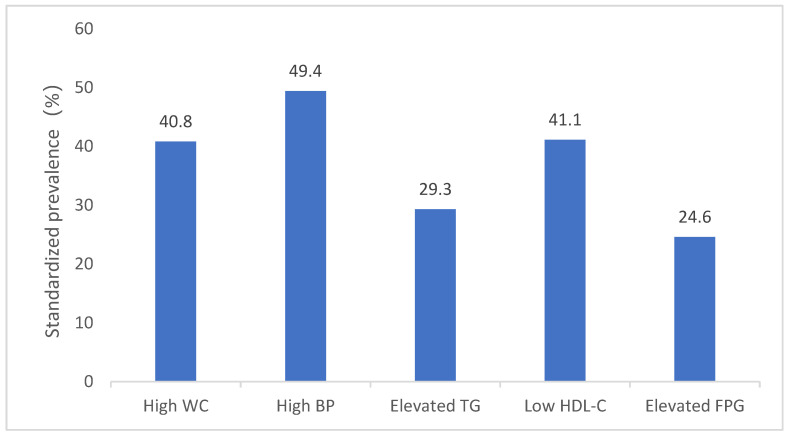
Standardised prevalence of metabolic components among adults in China. WC, waist circumference; BP, blood pressure; TG, triglycerides; HDL-C, high-density lipoprotein cholesterol; FPG, fasting plasma glucose.

**Figure 2 nutrients-13-04475-f002:**
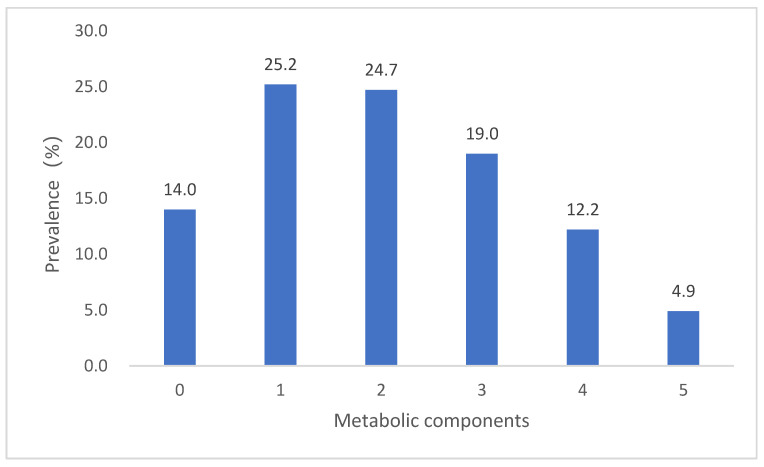
Metabolic components among adults in China.

**Table 1 nutrients-13-04475-t001:** Participant characteristics according to sex.

	Men	Women	Total
N	61,775	68,243	130,018
Age, years *	53.0 (43.4,63.2)	51.8 (42.4,61.7)	52.3 (42.9,62.4)
Residence location, N (%) *			
City	25,371 (41.1)	29,878 (43.8)	55,249 (42.5)
Countryside	36,404 (58.9)	38,365 (56.2)	74,769 (57.5)
Area of the country, N (%) *			
South	31,232 (50.6)	35,253 (51.7)	66,485 (51.1)
North	30,543 (49.4)	32,990 (48.3)	63,533 (48.9)
Education level, N (%) *			
Low	24,315 (39.4)	36,645 (53.7)	60,960 (46.9)
Moderate	22,516 (36.5)	18,608 (27.3)	41,124 (31.6)
High	14,944 (24.2)	12,990 (19.0)	27,934 (21.5)
Income, N (%) *			
Low	17,612 (28.5)	18,535 (27.2)	36,147 (27.8)
Moderate	15,524 (25.1)	17,265 (25.3)	32,789 (25.2)
High	15,715 (25.4)	17,995 (26.4)	33,710 (25.9)
Very high	12,924 (20.9)	14,448 (21.2)	27,372 (21.1)
Family history of hypertension, N (%) *	20,313 (32.9)	23,619 (34.6)	43,932 (33.8)
Family history of diabetes, N (%) *	5609 (9.1)	7149 (10.5)	12,758 (9.8)
Smoking, N (%) *			
Never	20,433 (33.1)	65,732 (96.3)	86,165 (66.3)
Former	8673 (14.0)	569 (0.8)	9242 (7.1)
Current	32,669 (52.9)	1942 (2.9)	34,611 (26.6)
Alcohol consumption, N (%) *			
Never	21,429 (34.7)	51,799 (75.9)	73,228 (56.3)
Moderate	28,415 (46.0)	15,377 (22.5)	43,792 (33.7)
Excessive	11,931 (19.3)	1067 (1.6)	12,998 (10.0)
Physically active, N (%) *	41,112 (66.6)	50,741 (74.4)	91,853 (70.7)
Insufficient fruit and vegetable intake, N (%) *	32,012 (51.8)	34,970 (51.2)	66,982 (51.5)
Red meat intake, N (%) *			
Moderate	4606 (7.5)	5714 (8.4)	10,320 (7.9)
Insufficient	24,442 (39.6)	31,533 (46.2)	55,975 (43.1)
Excessive	32,727 (53.0)	30,996 (45.4)	63,723 (49.0)
BMI (kg/m²) *	24.2 (22.0,26.6)	24.2 (22.0,26.7)	24.2 (22.0,26.6)
WC (cm) *	85.3 ± 10.0	81.9 ± 9.7	83.5 ± 10.0
Systolic blood pressure (mmHg) *	136.6 ± 19.7	134.4 ± 22.2	135.4 ± 21.0
Diastolic blood pressure (mmHg) *	81.7 ± 11.4	77.7 ± 11.4	79.6 ± 11.5
TG (mmol/L) *	1.3 (0.9,2.0)	1.2 (0.8,1.8)	1.2 (0.8,1.9)
HDL-C (mmol/L) *	1.2 ± 0.4	1.3 ± 0.3	1.3 ± 0.3
Fasting plasma glucose (mmol/L) *	5.2 (4.8,5.7)	5.2 (4.8,5.6)	5.2 (4.8,5.7)

Data are presented as $\bar{x}$ ± s for normal distributions, or as M (P25, P75) for skewed distributions, or N (%). * *p* < 0.05 compared with men. Abbreviations: BMI, body mass index; WC, waist circumference; TG, triglycerides; HDL-C, high-density lipoprotein cholesterol.

**Table 2 nutrients-13-04475-t002:** Prevalence of metabolic syndrome in adults according to participant characteristics.

		Prevalence	95% CI	Rao-Scott X^2^	*p*-Value
Total		31.1	30.0–32.2		
Sex				10.9707	0.0009
	Men	30.0	28.8–31.3		
	Women	32.3	30.9–33.6		
Age, years				1420.8214	<0.0001
	20–44	23.3	22.1–24.5		
	45–59	39.0	37.9–40.1		
	60–74	43.9	42.4–45.5		
	≥75	44.2	41.3–47.2		
Residence location				2.8593	0.0908
	City	32.0	30.2–33.8		
	Countryside	30.1	28.8–31.4		
Area of the country				232.4353	<0.0001
	South	27.4	26.2–28.6		
	North	35.9	34.7–37.2		
BMI				54,564.5498	<0.0001
	Normal	11.8	11.1–12.6		
	Overweight	41.0	39.9–42.0		
	Obese	70.5	69.0–72.0		
Education level				59.5475	<0.0001
	Low	35.8	34.4–37.2		
	Moderate	30.3	29.0–31.6		
	High	27.1	24.9–29.3		
Income				1.5158	0.6786
	Low	30.9	29.5–32.4		
	Moderate	30.4	29.2–31.6		
	High	31.7	30.4–33.1		
	Very high	31.4	28.7–34.1		
Family history of hypertension				87.8033	<0.0001
	No	29.0	28.0–30.0		
	Yes	34.8	33.1–36.4		
Family history of diabetes				107.2048	<0.0001
	No	30.0	28.9–31.0		
	Yes	39.9	37.5–42.3		
Physical activity				6.7119	0.0096
	Not active	32.1	30.7–33.5		
	Active	30.6	29.5–31.8		
Smoking status				34.7917	<0.0001
	Never	31.2	30.0–32.5		
	Former	37.7	35.4–40.0		
	Current	29.6	28.1–31.0		
Alcohol consumption				40.8781	<0.0001
	Never	32.6	31.6–33.7		
	Moderate	28.7	27.2–30.2		
	Excessive	33.2	30.9–35.5		
Fruit and vegetable intake				2.3882	0.1223
	Sufficient	31.6	30.5–32.7		
	Insufficient	30.6	29.2–32.0		
Red meat intake				36.6381	<0.0001
	Moderate	30.8	29.0–32.5		
	Insufficient	32.8	31.5–34.1		
	Excessive	29.8	28.6–31.0		

Abbreviations: BMI, body mass index.

**Table 3 nutrients-13-04475-t003:** Logistic regression analysis results of influencing factors.

Influencing Factor		β	SE	Wald X^2^	*p*	OR	95% CI
Intercept		−12.4553	0.0961	16,784.5555	<0.0001		
Sex	Women vs. Men	0.5727	0.0189	918.6448	<0.0001	1.773	1.709–1.840
Age		0.0368	0.0005	4751.4066	<0.0001	1.037	1.036–1.039
Residence location	Countryside vs. City	−0.0411	0.0148	7.7242	0.0054	0.960	0.932–0.988
Area of the country	North vs. South	0.0835	0.0136	37.4822	<0.0001	1.087	1.058–1.117
BMI		0.3377	0.0024	19,715.8134	<0.0001	1.402	1.395–1.408
Income	Moderate vs. Low	0.0433	0.0188	5.3170	0.0211	1.044	1.007–1.083
	High vs. Low	0.0800	0.0190	17.6248	<0.0001	1.083	1.044–1.124
	Very high vs. Low	0.1160	0.0208	30.9902	<0.0001	1.123	1.078–1.170
Family history of hypertension	Yes vs. No	0.2130	0.0146	212.8383	<0.0001	1.237	1.203–1.273
Family history of diabetes	Yes vs. No	0.3992	0.0225	315.9245	<0.0001	1.491	1.426–1.558
Smoking status	Former vs. Never	0.0289	0.0292	0.9808	0.3220	1.029	0.972–1.090
	Current vs. Never	0.1339	0.0207	41.6962	<0.0001	1.143	1.098–1.191
Alcohol consumption	Moderate vs. Never	−0.0868	0.0160	29.6303	<0.0001	0.917	0.889–0.946
	Excessive vs. Never	0.0001	0.0254	0.0000	0.9957	1.000	0.952–1.051
Physical activity	Active vs. Not active	−0.1197	0.0149	64.7071	<0.0001	0.887	0.862–0.913

Abbreviations: BMI, body mass index; CI, confidence interval; OR, odds ratio; SE, standard error.

## Data Availability

The data is not allowed to be disclosed according to the National Institute for Nutrition and Health, Chinese Center for Disease Control and Prevention.

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
