# Peer review of "Prevalence and Influencing Factors of Metabolic Syndrome among Adults in China from 2015 to 2017"

_nutrients, 2021, doi:10.3390/nu13124475_

Round 1
Reviewer 1 Report
The article is well structured and provides data of considerable interest. However, there are formal flaws that need to be corrected.
In section 2.5 (covariates) indiscriminate use of upper and lower case letters is used. For example: Smoking was divided into Never smoking, Former smoking and Current smoking; Drinking was divided into never drinking, moderate drinking (men drink less than 25g and women drink less than 15g per day) and excessive drinking.
In lines 113 and 114 the quotations 8]; (8) and [9]; (9) are repeated.
In line 70 (Using TZG height and sitting height) explain what TZG means.
In line 119. What does asx±s stand for mean and standard deviation?
In table 1, differences by sex are presented, while in table 2 and 3, gender is mentioned. Please unify the term. In my opinion, we are working on the biological sex here, as the gender with which the subjects identify themselves is unknown.
The tables should be self-explanatory, so I recommend that in the column headings you indicate whether mean, median and 25th and 75th percentile values etc. are presented.
The bibliography also contains several errors. Citations 8 and 9 are not complete. In number 8 the year of publication is missing. Citation 9 is as indicated: World Health Assembly, 66. Draft action plan for the prevention and control of noncommunicable diseases 2013-2020: report by the Secretariat. World Health Organization. 2013 https://apps.who.int/iris/handle/10665/105668
In citation 20, the names of all authors are in capital letters.
Some citations e.g. 22 and 23 have the journal title in acronym format, while most are in full title. Please unify according to Nutrients standards.
Regarding the content of the article, you consider the range of normopese, overweight and obesity according to a Chinese standard which you should explain. Citation 2 corresponds to an article where this aspect is not mentioned. This is important because readers expect you to have used the criteria of the World Health Organisation, which considers overweight from 25 kg/m2 and obesity from 30 kg/m2.
Author Response
Thank you for your suggestions. I have made relevant modifications, which are as follows:
- Question: In section 2.5 (covariates) indiscriminate use of upper and lower case letters is used. For example: Smoking was divided into Never smoking, Former smoking and Current smoking; Drinking was divided into never drinking, moderate drinking (men drink less than 25g and women drink less than 15g per day) and excessive drinking.
Answer: Thank you, the relevant parts have been modified.
- Question: In lines 113 and 114 the quotations 8]; (8) and [9]; (9) are repeated.
Answer: Thanks, but [8]; (8) and [9]; (9) are not repeated. [8]/[9] refers to references, (8)/(9) is the classification number of covariates.
- Question: In line 70 (Using TZG height and sitting height) explain what TZG means.
Answer: Thank you, TZG is a type of height-sitting meter produced by China Wuxi Weighing Instrument Factory Co., Ltd.
- Question: In line 119. What does asx±s stand for mean and standard deviation?
Answer: Thank you, the relevant parts have been modified.
- Question: In table 1, differences by sex are presented, while in table 2 and 3, gender is mentioned. Please unify the term. In my opinion, we are working on the biological sex here, as the gender with which the subjects identify themselves is unknown.
Answer: Thanks, it has been unified into sex.
- Question: The tables should be self-explanatory, so I recommend that in the column headings you indicate whether mean, median and 25th and 75th percentile values etc. are presented.
Answer: Thank you, the relevant explanation has been added below table 1.
- Question: The bibliography also contains several errors. Citations 8 and 9 are not complete. In number 8 the year of publication is missing. Citation 9 is as indicated: World Health Assembly, 66. Draft action plan for the prevention and control of noncommunicable diseases 2013-2020: report by the Secretariat. World Health Organization. 2013 https://apps.who.int/iris/handle/10665/105668
In citation 20, the names of all authors are in capital letters.
Some citations e.g. 22 and 23 have the journal title in acronym format, while most are in full title. Please unify according to Nutrients standards.
Answer: Thank you, the format of references has been modified.
- Question: Regarding the content of the article, you consider the range of normopese, overweight and obesity according to a Chinese standard which you should explain. Citation 2 corresponds to an article where this aspect is not mentioned. This is important because readers expect you to have used the criteria of the World Health Organisation, which considers overweight from 25 kg/m2 and obesity from 30 kg/m2.
Answer: Thank you for your advice. ①(2) is the classification number of covariates, not refers to references. ②As the object of this study is Chinese, our research team think using Chinese standards to judge whether overweight and obesity is more convincing.
It should be noted: I invited relevant experts to modify English. Because of a large number of revisions, there is no marked revision of the manuscript. I'm sorry for the inconvenience. I hope you can understand.
Thanks again for your help.

Reviewer 2 Report
In this cross-sectional study conducted in 130018 subjects from 31 Chinese provinces, aged 20 and above , the authors showed a higher prevalence of MetS and MetS parameters in this populations with a significantly higher prevalence in women than men and in north than south population.
This paper deals with an interesting topic. However, I have some concerns.
Introduction, materials and methods must be improve. More information about the study should be given. For instance what kind of dietary and physical activity questionnaire were used into the study?
As recognized by the authors, the major limitation is the cross-sectional analyses who do not allow to draw any conclusions about mechanism/cause effect relationship. However the manuscript does not provide any mechanistic insight into how they may be related for instance MetS and gender differences.
Despite the large population sample and the well-conducted analysis, I think this article provides minimal incremental knowledges.
Further, English should be strongly revised.
Author Response
Thank you for your suggestions. I have made relevant modifications, which are as follows:
- Question: Introduction, materials and methods must be improve. More information about the study should be given.
Answer: Thank you, the summary has been carefully modified.
- Question: For instance what kind of dietary and physical activity questionnaire were used into the study?
Answer: Thanks. ①Dietary use the national unified food frequency questionnaire to collect the frequency and consumption of various foods of the respondents aged 18 and over in the past year. ②Use global physical activity question naire for physical activities.
- Question: As recognized by the authors, the major limitation is the cross-sectional analyses who do not allow to draw any conclusions about mechanism/cause effect relationship. However the manuscript does not provide any mechanistic insight into how they may be related for instance MetS and gender differences.
Answer: Thank you, the discussion section has been added a little mechanistic insight into gender differences.
- Question: Despite the large population sample and the well-conducted analysis, I think this article provides minimal incremental knowledges.
Answer: Thank you, this study can well reflects the epidemic characteristics of MetS among residents aged 20 years and older in China, and there is no recent nationwide survey on the prevalence of MetS in China. The results of this study are intended to be the next research direction.
- Question: Further, English should be strongly revised.
Answer: Thanks, English has been revised.
It should be noted: I invited relevant experts to modify English. Because of a large number of revisions, there is no marked revision of the manuscript. I'm sorry for the inconvenience. I hope you can understand.
Thanks again for your help.
